# Fault Detection in Induction Machines Using Learning Models and Fourier Spectrum Image Analysis

**DOI:** 10.3390/s25020471

**Published:** 2025-01-15

**Authors:** Kevin Barrera-Llanga, Jordi Burriel-Valencia, Angel Sapena-Bano, Javier Martinez-Roman

**Affiliations:** Institute for Energy Engineering, Universitat Politècnica de València, Camino. de Vera s/n, 46022 Valencia, Spain; kebarlla@upv.edu.es (K.B.-L.); jorburva@die.upv.es (J.B.-V.); asapena@die.upv.es (A.S.-B.)

**Keywords:** fault diagnosis, induction motors, spectral images, deep learning, explainability, predictive maintenance

## Abstract

Induction motors are essential components in industry due to their efficiency and cost-effectiveness. This study presents an innovative methodology for automatic fault detection by analyzing images generated from the Fourier spectra of current signals using deep learning techniques. A new preprocessing technique incorporating a distinctive background to enhance spectral feature learning is proposed, enabling the detection of four types of faults: healthy motor coupled to a generator with a broken bar (HGB), broken rotor bar (BRB), race bearing fault (RBF), and bearing ball fault (BBF). The dataset was generated from three-phase signals of an induction motor controlled by a Direct Torque Controller under various operating conditions (20–1500 rpm with 0–100% load), resulting in 4251 images. The model, based on a Visual Geometry Group (VGG) architecture with 19 layers, achieved an overall accuracy of 98%, with specific accuracies of 99% for RAF, 100% for BRB, 100% for RBF, and 95% for BBF. A new model interpretability was assessed using explainability techniques, which allowed for the identification of specific learning patterns. This analysis introduces a new approach by demonstrating how different convolutional blocks capture particular features: the first convolutional block captures signal shape, while the second identifies background features. Additionally, distinct convolutional layers were associated with each fault type: layer 9 for RAF, layer 13 for BRB, layer 16 for RBF, and layer 14 for BBF. This methodology offers a scalable solution for predictive maintenance in induction motors, effectively combining signal processing, computer vision, and explainability techniques.

## 1. Introduction

Induction motors (IM) have a significant role in modern industry, representing approximately 80% of rotating machinery in industrial applications. Their popularity is due to features such as energy efficiency, low maintenance costs, and adaptability to different environments and operational demands [1]. These motors are used across various industrial sectors, including manufacturing, power generation, transportation, and construction, where they are responsible for driving pumps, fans, compressors, and various production equipment. The reliability of induction motors is essential to ensure continuous operation and to avoid costly interruptions in industrial processes [2].

However, like all mechanical equipment, induction motors are subject to various types of faults that can arise due to operating conditions, natural wear, or manufacturing defects, which affect their performance and reduce their lifespan [3]. The most common rotor faults, such as broken or asymmetric bars, usually occur due to overloads, material fatigue, or issues in the manufacturing process, affecting the symmetry of rotation and causing instability in motor operation [4]. Similarly, bearing problems, whether in the races or the rolling elements, may result from inadequate lubrication, misalignment, or contamination by external particles, leading to vibrations and abnormal noises that accelerate component wear. On the other hand, the most common faults in the stator generally occur due to insulation degradation of the windings or overheating, as well as conditions that increase the risk of short circuits and reduce efficiency in energy conversion [5].

These faults, if not identified and addressed in time, can evolve into catastrophic failures, resulting in costly production stoppages, damage to related equipment, maintenance or replacement expenses, and significant economic losses [6].

In response to this issue, predictive maintenance has emerged as a strategy to monitor the conditions of induction motors. This approach relies on continuous equipment monitoring to detect anomalies or faults at early stages, allowing maintenance interventions to be scheduled before critical failures occur [7]. Among the various monitoring techniques available, Motor Current Signature Analysis (MCSA) has established itself as an effective and noninvasive method for fault detection, which is characterized by its ease of implementation, minimal technical requirements, and its ability to identify various types of faults simultaneously [8,9].

With the advent of Industry 5.0 and the rise of artificial intelligence, machine learning methods have gained importance in fault detection for induction motors. Classical machine learning techniques, such as Support Vector Machine (SVM) [10], Decision Trees [11], and Artificial Neural Networks [12], have proven effective in classifying different types of faults through features extracted from motor current signals. However, these supervised methods often rely on manual and labor-intensive processes to select and extract relevant features, limiting their ability to capture complex and nonlinear patterns in induction motor data. This has driven the exploration of deep learning techniques, which can automatically identify patterns without the need for feature preselection.

In this context, the Recurrent Neural Network (RNN) [13] and is variants such as Long Short-Term Memory (LSTM) [14], have been successfully used in time series analysis, applying current signals to capture the dynamics and temporal dependencies that reflect the motor’s condition. More recently, Transformer models [15] have demonstrated advantages over the traditional RNN, using attention mechanisms to capture long-term relationships in time series data, which make them particularly useful in scenarios with extended sequence dependencies.

Parallel to these deep learning approaches based on the direct analysis of raw current signals, the visual analysis of signal behavior has been of importance to experts in terms of predictive maintenance of induction motors [16]. These experts often closely observe patterns and characteristics present in the Fourier spectrum of current signals, where specific harmonics and patterns indicate the presence of particular types of faults [17]. This approach is relevant in industrial environments, where varying operating conditions can complicate manual analysis and increase the value of automatic pattern recognition.

Inspired by this visual approach of experts and based on artificial intelligence’s ability to analyze images, Convolutional Neural Networks (CNN) [18] have been implemented to emulate and automate this type of visual analysis. CNN can learn to identify complex visual patterns in images, capturing spatial features that may be relevant in fault diagnosis. In this way, CNNs serve as a complementary tool for experts, automating visual analysis and providing a new layer of interpretation that enriches the understanding of spectral data, thus supporting and improving fault diagnosis in induction motors [19].

In recent years, researchers have explored the possibility of converting current signals from induction motors into visual representations, with the aim of harnessing the potential of deep learning models in image analysis. However, most of these approaches have focused on directly analyzing the signal waveform or encoding it in two dimensions, without adequately considering the importance of preserving and highlighting the specific spectral features that distinguish each type of fault [20]. Accurately representing these spectral patterns can improve the accuracy of automatic fault diagnosis.

Furthermore, a challenge in applying deep learning models to this field is the interpretability of the results. These complex models are often perceived as black boxes that provide output without offering a clear explanation about the patterns or features that drive the prediction [21]. Incorporating explainability techniques makes it possible to verify the accuracy of the model, understand the basis for its decisions, detect potential biases, and improve user confidence in the tool. Explainability facilitates the identification of the image regions that contribute to each diagnosis, helping predictive maintenance experts interpret and validate the model’s behavior under different engine operating conditions [22].

Starting from the expert visual approach and taking advantage of the potential of artificial intelligence in image analysis, this work proposes a methodology that combines advanced signal processing and deep learning techniques for automatic fault detection in induction motors. The main objective is to develop a system capable of detecting and classifying different types of faults, as well as provide interpretive information on the model’s decision process. This methodology is applied to the detection and classification of four specific fault types. It is designed to be part of a two-stage system where an initial fast classifier identifies healthy or unhealthy conditions. Once unhealthy conditions are detected, this system provides a detailed classification of faults, making it well suited for integration into routine operations and predictive maintenance strategies. In particular, this study presents the following contributions:A new preprocessing technique for images generated from the Fourier spectra of current signals, incorporating a distinctive background designed to enhance the model’s ability to learn the relative position of spectral peaks and their amplitudes.A deep learning model based on the VGG19 architecture, which is configured to accurately classify four common types of faults in induction motors: Healthy motor coupled to a generator with a broken bar (HGB), Broken Rotor Bar (BRB), Race Bearing Fault (RBF), and Bearing Ball Fault (BBF).An analysis of the model’s interpretability using advanced explainability techniques to identify the visual and spectral features that the model uses in its classification process.A validation of the proposed system under various operating conditions, demonstrating its applicability in industrial settings.

### Literature Review

MCSA techniques represent a traditional and widely used approach to monitoring and detecting faults in induction motors. These techniques have proven effective in accurately estimating speed in induction motors, both in sensor-based and sensorless systems, as well as in detecting specific faults, such as short circuits in stator windings. However, MCSA performance can be conditioned by variations in the operating conditions [23]. In parallel, spectral analysis using Fourier transforms on current signals has been an important tool for predictive maintenance experts. Spectrum patterns and characteristics allow for identifying specific types of faults [9], providing a noninvasive diagnosis based on the characteristic frequencies of common rotor and bearing faults.

While MCSA and spectral analysis have proven to be effective methods, both can benefit from integration with machine learning techniques. The use of the SVM was one of the first approaches in this direction, allowing for the development of models capable of identifying relationships between current amplitudes and fault types [24]. SVM were initially chosen for their low computational cost, although their accuracy is limited in highly complex scenarios. Considering the current signal as a time series, the use of RNN allows for modeling sequential dependencies in the data, making them useful for the extraction of temporal features [25] and for the detection of faults in induction motors [26]. However, when data sequences are long, RNN have difficulties in retaining past information, which is known as the out-of-memory problem. In response, LSTM networks have been implemented as an improvement to reduce this error and have shown good performance in diagnosing broken bar faults in induction motors [27].

More recently, techniques based on attention mechanisms have emerged that allow for assigning an importance weight to current values over time, thus relating nonsequential information in time series. This mechanism, combined with RNN, has shown potential in detecting short-circuit faults in synchronous machines [28]. With the creation of multiple simultaneous attention heads and the incorporation of value encodings, the Transformer architecture emerged, which has been applied to capture multiple complex relationships in the current signals of induction motors, managing to identify fault patterns in the rotor [29]. Together, these direct current analysis techniques have proven effective in identifying fault patterns in induction motors. However, these approaches, while capturing temporal dependencies, often overlook specific spectral features observable through visual analysis, which can be important for identifying the subtle patterns associated with certain faults. Therefore, this direct analysis is frequently complemented by the visual examination of Fourier spectra, where specific frequency patterns provide information for fault identification.

Building upon this common practice of expert visual analysis, researchers have explored methods to transform current signals into visual representations suitable for automated processing. Techniques such as Fast Fourier Transform (FFT) [30], Wavelet transform [31], Wigner–Ville distribution [32], and Gabor analysis [33] are among the most widely used, showing promising results in representing the time–frequency characteristics of the signals [34]. These transformations have proven especially effective when combined with computer vision models, particularly CNNs, due to their ability to capture spatial patterns in complex visual data.

The evolution of these approaches has been gradual and systematic. Early approaches used 1D CNN models to process current signals directly in one dimension, focusing primarily on fault detection in induction motor bearings [35,36]. Subsequently, research moved towards the use of two-dimensional CNN, successfully predicting the number of broken rotor bars by analyzing two-dimensional current–angular position images [37]. Further studies have demonstrated the effectiveness of these models in detecting faults in stator [38] and three-phase inverters under different load conditions [39] using direct representations of the current signals and their Fourier transforms. Furthermore, hybrid architectures combining the CNN with LSTM have been explored to improve the accuracy of bearing fault detection [40].

However, a common limitation in these previous studies is that they have mainly focused on the direct implementation of CNNs on signals transformed into images, without considering two fundamental aspects: the preservation of relevant spectral information in the visual representation and the interpretability of the classification process. This spectral information is important, as it contains frequency patterns that are directly associated with specific types of faults. Thus, although these models have demonstrated good results in terms of accuracy, they operate as black boxes where the basis of their decisions remains hidden, and they may lose spectral information during the signal-to-image transformation process.

The present work proposes an innovative approach that addresses these limitations through two main contributions. First, it introduces a new preprocessing technique for images generated from Fourier spectra that incorporates a distinctive background specifically designed to preserve and highlight the spectral features associated with each fault type, making it easier for the CNN to better understand the relationship between the signal, frequencies, and magnitude in decibels. Second, it implements explainability techniques that allow for understanding which specific features of the Fourier spectrum are relevant for the identification of each fault type by the CNN.

This paper is organized as follows: Section 2 describes the proposed method. Section 2.2 presents the dataset used for model learning, covering the operating conditions considered and the signal preprocessing process. Section 2.3 describes the model selection, including the architecture used and the specific CNN configurations. Section 2.4 indicates the classification model training process, the evaluation metrics, the selected hyperparameters, and the validation steps. Section 3 evaluates the model on a test set under different operating conditions, analyzing its performance using classification metrics. Section 4.3 examines the interpretability of the model by analyzing it externally, using local explainability techniques to identify image regions that contribute to fault predictions, and internally through an in-depth examination of the convolutional layers. Finally, Section 5 presents the conclusions and highlights the contributions of the work, as well as possible future directions for research.

## 2. Methodology

### 2.1. Overview

This section describes the proposed methodology for the automatic detection and classification of faults in induction motors, leveraging signal processing and deep learning techniques. The overview of the proposed system is illustrated in Figure 1.

The process begins by applying the FFT to the current signal of a three-phase induction motor to obtain the frequency spectrum (Figure 1a). Instead of using this spectrum directly as an input image, a custom-designed background is introduced (Figure 1b), which is then combined with the FFT data (Figure 1c). The background is specifically designed to segment frequency ranges and highlight spectral amplitudes associated with each fault type.

The resulting enhanced FFT representation (Figure 1d) is resized to a standard pixel size of 224 × 224 (Figure 1e) to serve as input to a CNN-based classification model (Figure 1f). This CNN model is trained to automatically classify the four fault conditions—HGB, BRB, RBF, and BBF—based on the visual information contained in the enhanced FFT representation.

In this model, the resized images are processed through convolutional layers, where filters sweep across the image to extract relevant features, generating feature maps. The dimensionality of these feature maps is subsequently reduced using pooling layers, which retain only the most significant features. This process is repeated across multiple convolutional layers, progressively abstracting and deepening the extracted features. The final feature maps are then flattened and connected to a classification stage composed of fully connected layers, which output probabilities for the four fault conditions. The fault type with the highest probability is considered the predicted class.

Finally, explainability techniques are employed to interpret the decision-making process of the CNN, identifying specific image regions, convolutional layers, and frequency components that contribute to the classification of each fault type.

### 2.2. Dataset

#### 2.2.1. Data Collection

The dataset used in this study was collected from an experimental test bench designed to emulate various operating conditions and faults in three-phase induction motors. The system included a three-phase induction motor coupled to a generator, with its speed controlled by an M4BP 160 MLB 3GBP drive operating under a Direct Torque Control (DTC) scheme. The induction motor and generator used in the setup have the following technical specifications: a rated voltage of 400 V, input current of 27.8 A, power output of 15 kW, operating frequency of 50 Hz, torque of 97.1 Nm, and a nominal speed of 1474 rpm.

The faults were physically implemented on the test bench using techniques designed to emulate real operating conditions. For bearing faults, controlled defects such as punctures of up to 6 mm were introduced into the bearing structure to simulate progressive mechanical damage. Additionally, faults in the bearing balls were replicated by creating localized deformations or wear. Regarding the rotor, faults were emulated by progressively removing the bars, representing broken bar scenarios. Furthermore, a healthy motor coupled to a generator with a broken bar was configured to analyze the interaction between both components and evaluate the impact of the fault under joint operating conditions. These modifications allowed for studying the motor’s response under different levels of fault severity.

The motor was operated on a test bench composed of a three-phase induction motor coupled to a generator controlled by a variable speed drive. The operating conditions included constant and variable speeds (such as acceleration and deceleration ramps), as well as different load levels.

Current signals from the three motor phases (u, v, and w) were recorded at four different speeds (20 rpm, 900 rpm, 1200 rpm, and 1500 rpm) and five different load levels (0%, 25%, 50%, 75%, and 100%). The data were collected under constant speed and load conditions for each combination. The signals were sampled at a frequency of 20 kHz, resulting in a total of 2,000,000 samples equivalent to 100 s of continuous data for each combination of speed and load.

During the preprocessing phase, the 100-s current signals were divided into 10-s segments, generating a total of 10 samples per phase for each combination of operating conditions. This resulted in a total of 30 samples for each speed and fault combination. The segmented signals were analyzed for potential spikes and outliers [41], which are abrupt deviations in amplitude that can distort the frequency analysis. Spikes typically occur due to transient disturbances, and outliers represent extreme values that deviate significantly from the expected range of the signal. No spikes or outliers were detected during this analysis.

Subsequently, the Fourier spectra were obtained by applying the FFT to the current signal segments. Each 10-s signal segment used for the FFT consisted of 200,000 samples derived from a sampling frequency of 20 kHz (N=10s×20,000samples/second=200,000samples). The FFT transforms the time-domain signal x(t) into its frequency-domain representation X(f), as defined in [42]:$$X\left(f\right)=\sum_{n=0}^{N-1}x\left(n\right)e^{-j2\pi fn/N}$$
where x(n) represents the sampled time-domain signal, *N* is the total number of samples, and *f* is the frequency index. A representation of the spectra for each fault type is shown in Figure 2. While the spectra are displayed up to 180 Hz for visualization purposes, the proposed model was designed to analyze frequencies up to 100 Hz.

#### 2.2.2. Preprocessing Techniques

Once the dataset was collected and the FFT signals were obtained, various preprocessing techniques were implemented to transform the raw data into visually interpretable representations. Preprocessing techniques, dataset partitioning, and model selection were simultaneously initiated. To facilitate their explanation, these stages are described separately, starting with the preprocessing techniques.

Preprocessing of the signals and their transformation into visual representations are the main contribution to the development of the proposed model. This process evolved through multiple stages, with each aimed at improving the quality and representability of the generated images. The different techniques tested and the reasoning behind each are described below.

Initially, the obtained FFT signals were used without additional modifications, converting them directly into visual representations that served as input to the model. For this stage, image encoding techniques such as the Gramian Angular Field (GAF) [43] and Markov Transition Field (MTF) [44] were explored, both of which are widely used for transforming time series data into images. Although these techniques allowed the signal to be represented in a structured manner, their effectiveness in capturing the specific spectral patterns associated with each fault type was limited, as they did not emphasize spectral features in terms of magnitude and frequency.

Subsequently, FFT signals with smoothed representations were used to improve the visual representation. To achieve this, the Savitzky–Golay filter [45] was applied, which is a technique that fits local polynomials of degree p=3 over 2m+1=51 neighboring points to smooth data without losing features. Using a degree-3 filter and a window of 51 points reduced irregularities while preserving the model’s ability to identify patterns in the images by minimizing noise and maintaining signal characteristics. The filter was implemented using the scipy.signal library in Python. The function takes as input the FFT vector, the window size (51 points), and the polynomial degree (3), returning the smoothed signal directly. The equation used to smooth the signal is as follows:$$y_{i}=\sum_{j=-m}^{m}c_{j}x_{i+j},$$
where $c_{j}$ = polynomial fitting coefficients, $y_{i}$ represents the smoothed values, $x_{i+j}$ represents the original data points, and $c_{j}$ represents the coefficients calculated based on the polynomial degree p=3 and window size 2m+1=51 of the Savitzky–Golay filter.

Next, visual modifications were made to the image background to provide additional references that would help the model interpret the relative position and amplitude of the spectral peaks. Initially, two reference lines were added: a horizontal line from 0 to −10 dB to highlight the distance of the peaks to the origin and a vertical line between 48 and 52 Hz to mark the peak corresponding to the motor frequency. These lines, differentiated by colors, proved useful for improving the model’s understanding of fundamental spectral features.

To better capture the distribution of amplitudes concerning frequency, horizontal stripes were added to the background, dividing the magnitudes into ranges: from −10 to −20 dB, −20 to −40 dB, −40 to −60 dB, and so on. These stripes helped highlight variations in the fault conditions, which resulted in a notable improvement in classification. Subsequently, vertical stripes were added every 10 Hz, enabling the network to more accurately identify the position of the spectral peaks. This combination of horizontal and vertical stripes was referred to as a degraded background, as it provided a grid of differentiated tones that segmented the spectrum according to the magnitude–frequency relationship.

Finally, the signal line in the image was reduced to a thickness of 0.5 pt to further emphasize features without overloading the visual representation. The final result was an FFT image smoothed using the Savitzky–Golay filter, with a degraded background composed of grids that delimited specific regions of the spectrum. The resulting image was then resized to a standard size of 224×224 pixels, as this is a common input size for pretrained CNN architectures, ensuring compatibility with the model. This transformation process is illustrated in Figure 3.

#### 2.2.3. Image Dataset

All signals processed following the techniques described in Section 2.2.2 were transformed into FFT images, obtaining a total of 1266 images of the different speed-, load-, and failure-type configurations.

Deep learning models require distinct datasets for training and testing. To do so, systematic partitions of the dataset were performed, evaluating various configurations. The best split followed the Pareto rule (80/20) [46], which states that approximately 80% of the effects come from 20% of the causes. Applied to the context of machine learning, this rule suggests that a model can learn most representative patterns with a significant proportion of data (80%), while the remaining 20% is sufficient to evaluate its generalization ability [47]. During the training process, the model learns by adjusting its internal weights to minimize the loss function based on the data provided. However, this continuous fine-tuning can lead the model to overfit the training data, i.e., memorize specific patterns instead of generalizing correctly to unseen data [48]. For this reason, it was necessary to split the learning set into a training subset (80%) and a validation subset (20%). This split allowed for tuning model hyperparameters, such as the learning rate and the number of epochs, using metrics derived from the validation set. The partition into training, validation, and test sets is shown in Table 1.

To avoid bias during training, a proportional data balancing technique [49] was implemented. This method was chosen for its simplicity and effectiveness in adjusting the number of samples per class to ensure uniform representation. Unlike oversampling, which can lead to redundancy, or SMOTE—which generates synthetic samples—this technique redistributes samples by duplicating underrepresented classes and reducing overrepresented ones, maintaining the integrity of the original data. The technique is defined as$$n_{i}^{\prime }=\left\lceil N\;\cdot \;\frac{n_{i}}{\sum_{j=1}^{C}n_{j}}\right\rceil ,$$
where $n_{i}^{\prime }$ is the adjusted number of samples for class *i*, $n_{i}$ is the original number of samples, *C* is the total number of classes, and *N* is the desired total number of samples in the balanced set.

Applying this technique, the training set was adjusted to 202 samples per class, while the validation set was balanced with 51 samples per class. In the original dataset, underrepresented classes, such as RBF, were increased to match the target number of samples, while overrepresented classes, such as BRB, were reduced. Through iterative experiments, varying the sample distribution and measuring the model’s performance, the partition of 202 samples per class for training and 51 samples per class for validation was identified as the optimal balance. This ensured that all classes contributed equally to the learning process and prevented less-represented classes from disproportionately influencing the model. The initial and balanced values are presented in Table 1.

In order to successfully implement proportional balancing, data augmentation techniques [50] were applied to increase the number of samples in underrepresented classes, such as RBF, and achieve the target of 202 samples per class in the training set and 51 in the validation set.

In the initial phase, two techniques were evaluated: vertical flipping and rotation in 90-degree increments. Flipping proved to be a reliable augmentation technique, as the degraded colors in the background maintained the positional relationship between spectral features and their locations within the image. This enabled the network to correctly interpret image orientation, even after inversion. Similarly, rotation proved effective due to the reference line between 0 and −10 dB in the background, which helped maintain signal orientation, allowing the network to correctly identify spectral features regardless of rotation angle. Contrast and brightness adjustments emerged as complementary techniques. These modifications enhanced the visibility of subtle spectral patterns, making features more distinguishable and improving differentiation between fault types. Additionally, zooming was implemented to focus on specific spectrum areas, such as the main harmonics, providing the model with representations of each class. These augmentations enhanced the model’s ability to generalize to new data by exposing it to a broader spectrum of variations within each class.

### 2.3. Selection Model

In parallel with the definition of the dataset described in the previous sections, the model for classification was selected, iteratively refining the approach until optimal results were achieved.

The proposed approach was primarily based on computer vision techniques for fault detection through spectral images. Initially, traditional machine learning models such as SVM, Multilayer Perceptron (MLP), random forests, and gradient boosting were evaluated, along with sequential models such as the RNN and transformers, which have proven effective in time series analysis.

However, although these models are competent for direct signal processing, they exhibited limitations in capturing the visual dependencies present in the spectral representations generated in this work. Since the proposed preprocessing transforms signals into images, computer vision-based models, such as the CNN and Vision Transformers (ViT), emerged as more suitable options for identifying the complex and specific visual patterns associated with each fault type.

Several CNN architectures were evaluated, including models such as InceptionV4 [51], which is recognized for its inception modules that process visual information at multiple scales; SENet154 [52], which incorporates squeeze-and-excitation blocks; ResNet50 and ResNet154 [53], which feature residual connections with varying depths; VGG16 and VGG19 [54], which are distinguished by their sequential architecture with 16 and 19 convolutional layers, respectively; EfficientNetV2 [55], which is designed for enhanced efficiency; and ConvNext [56], which integrates modern architectural improvements. Additionally, ViT architectures were tested in small, base, and large configurations with a patch size of 16×16 [57]. Despite their recent success in various computer vision tasks, the ViT models demonstrated lower performance and required significantly greater computational resources.

All models were trained using pretrained weights from ImageNet [58] and transfer learning techniques. The training was conducted using the PyTorch framework on a 12 GB NVIDIA T4 GPU provided by the Google Colab environment. The training process, applied across all architectures, is detailed in the following Section 2.4. The comparative performance metrics for each tested architecture are presented in Table 2. VGG19 achieved the highest accuracy of 98.4% on the test set, standing out as the best performing architecture among those evaluated for this work. From an initial perspective, its success can be attributed to its deep yet simple design. In the rest of the manuscript, we will detail how this architecture was used for fault diagnosis, including an analysis of the convolutional layers and their relation to different fault types in Section 4.3.

### 2.4. Model Training

The VGG19 architecture is a CNN architecture that processes images through a series of convolutional blocks. Each block consists of convolutional layers followed by max pooling operations. The architecture, as given in Table 3, consists of 16 convolutional layers organized into five blocks, followed by a flattening layer and three fully connected layers. This structure allows for hierarchical feature extraction followed by a classification stage.

Transfer learning was used, where the pretrained weights of the model were used, and these were fine-tuned with the dataset described in the Section 2.2. The training process began with input images of size 224×224×3, where the three channels correspond to the RGB color space. In the first convolutional block, 64 filters were applied, producing feature maps of size 224×224×64. Through the subsequent blocks, the spatial dimensions were progressively reduced, while the number of filters was increased (128, 256, 512, and 512 filters in the respective blocks). After the convolutional blocks, the resulting feature maps were flattened into a vector of size 25,088, which was then processed through fully connected layers of sizes 4096 and 2048 before the final output layer of size 4, corresponding to the four fault types.

The training process implemented forward and backward propagation. During forward propagation, the input images passed through the network, generating predictions. Then, backward propagation computed the gradients of the loss function with respect to the network’s parameters, enabling parameter updates. The Adam optimizer was selected for its adaptive learning rate capabilities and efficient handling of sparse gradients. Since this is a multiclass classification problem, categorical cross-entropy was chosen as the loss function, which is defined as$$L=-\frac{1}{N}\sum_{i=1}^{N}\sum_{j=1}^{C}y_{ij}\log\left({\hat{y}}_{ij}\right),$$
where *N* is the total number of samples, *C* is the number of classes, $y_{ij}$ is the true value (1 if the sample belongs to class *j* and 0 if otherwise), and ${\hat{y}}_{ij}$ is the predicted probability for class *j*.

The training hyperparameters were selected through iterative testing. An initial learning rate of 0.001 was used, controlling the step size during parameter updates. The batch size was set to 32 images, meaning the training dataset was divided into groups of 32 images. The network processed these batches sequentially, updating the parameters after each batch. Once all batches had been processed (completing one epoch), the training loss was calculated. Additionally, the validation set was evaluated to measure validation accuracy, which represents the percentage of correctly classified samples in the validation set.

To optimize training, an early stopping technique was implemented based on monitoring the validation loss. The training process continued for a maximum of 81 epochs, tracking both the training and validation losses. The system was designed to terminate training if the validation loss did not show any improvement (threshold of 0.01) for five consecutive epochs, helping to prevent overfitting and ensuring model convergence. The goal was to maintain similar trajectories between the training and validation losses, as significant divergence could indicate issues of overfitting or underfitting.

Figure 4 illustrates the training and validation loss curves, along with the training accuracy throughout the training process. The convergence behavior of the model is observed in the loss curves, while the accuracy curve shows the performance improvement across epochs. At epoch 81, the system achieved an accuracy of 0.991 on the validation set.

## 3. Results of Classification Metrics

The objective of this section is to assess the model’s effectiveness in detecting the four types of faults using an unseen test dataset. The evaluation results are summarized in the confusion matrix shown in Table 4. In this matrix, rows correspond to the true classes, while columns represent the predicted classes. Each cell (i,j) indicates the proportion of samples from true class *i* that were classified as predicted class *j*. The values along the main diagonal represent the correctly classified samples for each class.

To quantify the model’s performance, classification evaluation metrics were calculated for each class based on the confusion matrix, including the precision, recall, and F1-score [59]. These metrics rely on the counts of True Positives (TP), False Positives (FP), True Negatives (TN), and False Negatives (FN). Specifically, TP represents the number of samples correctly predicted as belonging to a specific class, FP corresponds to samples incorrectly predicted as belonging to a class, TN includes samples correctly identified as not belonging to the class, and FN refers to samples of a class that the model failed to identify.

Precision measures the proportion of correct predictions for a specific class among all samples classified as belonging to that class. Recall evaluates the proportion of samples from a specific class correctly identified with respect to the total actual samples of that class. The F1-Score combines precision and recall into a single metric by calculating their harmonic mean. This allows for balancing both aspects [60].$$\mathrm{Precision}=\frac{\mathrm{TP}}{\mathrm{TP}+\mathrm{FP}}\;\;\mathrm{Recall}=\frac{\mathrm{TP}}{\mathrm{TP}+\mathrm{FN}}\;\;F1=2\;\cdot \;\frac{\mathrm{Precision}\;\cdot \;\mathrm{Recall}}{\mathrm{Precision}+\mathrm{Recall}}$$

Given that the test set was imbalanced, the Jaccard Index was employed as a metric that penalizes both false positives and false negatives, providing a more balanced evaluation of the model’s performance [61]. This metric assesses the similarity between the predictions and the actual labels, accounting for the imbalanced data. It is defined as follows:$$\mathrm{Jaccard}\;\mathrm{Index}=\frac{\mathrm{TP}}{\mathrm{TP}+\mathrm{FP}+\mathrm{FN}}$$

The results obtained for each class are summarized in Table 5. The model demonstrated strong performance across all metrics. For the HGB class, the model achieved a precision of 1, recall of 0.990, F1-score of 0.995, and Jaccard Index of 0.990. For the BRB class, a precision of 0.971, recall of 1.000, F1-score of 0.985, and Jaccard Index of 0.971 were observed. Similarly, for the RBF class, the model obtained a precision of 0.971, recall of 1.000, F1-score of 0.985, and Jaccard Index of 0.971. Finally, for the BBF class, the metrics achieved were a precision of 1.000, recall of 0.950, F1-score of 0.974, and Jaccard Index of 0.950.

The global metrics of the model, calculated as averages, reported an overall accuracy of 0.985, an average precision of 0.986, an average recall of 0.985, an average F1-score of 0.985, and an average Jaccard Index of 0.971.

### Performance Analysis Under Operating Conditions

To further analyze the model’s effectiveness, its performance was evaluated under different operating conditions, including load percentages and engine speeds. The test dataset was divided based on five distinct load conditions to evaluate the model’s accuracy across scenarios. Table 6 details the classification accuracy for each fault type at various load percentages.

The model consistently achieved high accuracy across all load conditions, with its accuracy remaining at 1.00 for most fault types. However, minor drops were observed for the BBF fault at 50% load (0.96) and for the HGB fault at 25% load (0.94).

The model’s accuracy under different motor speeds is presented in Table 7, showing the classification accuracy for each fault type at different operating speeds.

The results indicate that the model demonstrates high stability in its performance under different operating speeds. Slight variations in accuracy were observed: at 1500 rpm, the BBF fault accuracy decreased to 0.98, while at 900 rpm, the HGB fault accuracy dropped to 0.96. These variations are considered minor and did not affect the model’s overall performance.

## 4. Discussion

This study presents an automated system for detecting four types of faults in induction motors using spectral images of current signals as input. A distinctive feature of the approach is the incorporation of a differentiating background, which is designed to separate frequencies and decibel levels, providing additional visual context that enhances the model’s ability to discriminate between different faults. Among the various architectures evaluated, including CNN and ViT, the VGG19 architecture demonstrated superior performance and was selected as the main model for classification. After training and evaluation using a dataset split from the outset, the model achieved an overall accuracy of 98.5% on the test set. This performance was validated under various operational conditions of load and speed, demonstrating its consistency in fault detection.

The following discussion will analyze the main findings of the study, which is organized into the following sections: evaluation of the preprocessing technique’s effectiveness, interpretation of the obtained results, integration of local explainability techniques, analysis of the computational resources used, and limitations of the proposed approach for fault detection.

### 4.1. Impact of Spectral Preprocessing on Model Performance

The process of transforming current signals into images evolved iteratively, undergoing various stages of experimentation. Initially, a direct conversion of the signals into spectral images was implemented. However, preliminary results revealed significant limitations, with an overall accuracy of just 67%, indicating that this approach failed to adequately capture the spectral characteristics of the faults.

To address this shortcoming, advanced encoding techniques such as the GAF and MTF, widely used to transform time series data into visual representations, were explored. These techniques showed an improvement in validation set accuracy but failed to translate this improvement to the test set, achieving accuracy values of 71% and 70%, respectively. This performance prompted a re-evaluation of the initial approach.

An analysis of the activations of convolutional filters during training revealed that multiple filters exhibited null activation values. This phenomenon was attributed to the intermittent activation of filters, which responded only to spectral lines while remaining inactive in the extensive white background regions. Comparing this behavior with human interpretation of spectra highlighted a discrepancy. When an expert analyzes a spectrum, they not only consider outliers but also their position in the frequency–magnitude coordinates. This contextual reference, essential for human interpretation, was absent in the images directly processed by the model.

To address this limitation, a differentiated background was introduced to delineate specific regions associated with frequencies and amplitude levels in decibels. This modification provided the model with a positive positional reference, facilitating better interpretation of the spectral patterns. The impact of this improvement was evident in the increased activations of the convolutional filters, resulting in a test set accuracy of 92%. Additionally, this enhancement proved consistent in tests conducted under various load and speed conditions.

### 4.2. Training Analysis and Results Obtained

The process of model selection and optimization involved multiple stages of refinement to achieve the reported accuracy levels. Initially, a custom architecture was designed from scratch, but the results were unsatisfactory, prompting a transition to pretrained models. These models, with initial weights trained on ImageNet [58], provided a strong starting point due to their proven effectiveness in image classification tasks. The initial transfer learning approach [62] achieved an accuracy of 92%, indicating good performance but leaving room for improvement.

The decision to train the model from scratch, tailoring it exclusively to fault images, represented a step forward in performance improvement. This change increased the test set accuracy to 94%. Additionally, the integration of data augmentation techniques, such as rotations, mirroring, and zooming, further enhanced the performance, achieving an accuracy of 96%. This enrichment of the dataset enabled the model to generalize better under various operating conditions.

Final optimization was achieved through the implementation of an early stopping technique, which was designed to monitor loss differences between epochs. A threshold of 0.01 was set for loss variations between training and validation, with a limit of five consecutive epochs without improvement. This approach helped prevent overfitting and improved model convergence, culminating in a final test set accuracy of 98.5%. These results highlight the effectiveness of the iterative approach adopted for model refinement.

The results were analyzed in detail using the confusion matrix presented in Table 4, which showed high values across all classes, with a slight decrease in the BBF class, where the accuracy reached 95%. To explore this variation, the model was evaluated under different speeds and load conditions (Table 6 and Table 7). It was identified that the decrease occurred specifically in combinations of 1500 rpm and 50% load. This behavior was consistent even in previous tests with other trained models, indicating that the issue was not solely related to the selected architecture but potentially to characteristics of the dataset or spectral patterns in that specific combination.

The experimental data set used in this study were collected in a previous study on a test bench designed to evaluate failures in induction motors. For this work, only the experimental conditions that guaranteed complete coverage of constant load and speed combinations were selected. Although data under variable speed conditions were also available, they were not included due to the absence of samples at certain load percentages under these conditions, which would have made a comparative analysis between all loads difficult.

### 4.3. Model Interpretability Analysis

To better understand why the model makes certain predictions, interpretability techniques were used with the aim of analyzing the internal activations and patterns learned by the model for each type of failure.

Considering the model as a black box, where only the input images and the probabilities of belonging to the four failure classes are available as output, various visual explainability techniques were evaluated. Among the techniques considered were LIMEs [63], Integrated Gradients [64], and Saliency Maps [65]. These techniques highlight the regions of the image that most influence the model’s prediction.

During tests conducted with images from the test set, it was observed that Local Interpretable Model-Agnostic Explanations (LIME) and Integrated Gradients did not show consistent patterns among images of the same class, which hindered interpretation. In contrast, the Saliency Maps technique demonstrated greater coherence, identifying specific and consistent activation areas for each class. The Saliency Maps technique is based on slightly modifying each pixel of the input image, measuring the changes in the model’s predictions and calculating the gradients with respect to the image. Pixels whose alteration causes greater changes in the prediction are considered relevant and are highlighted in the resulting map. This procedure was applied separately to each class in the test set, allowing for the identification of highlighted areas based on the type of failure [66]. Figure 5 presents an example of the interpretability analysis. Row 1 shows FFT images corresponding to (A) HGB, (B) BRB, (C) RBF, and (D) BBF under operating conditions of 1500 rpm and 100% load. Row 2 of the figure displays the corresponding saliency maps for each image.

The analysis revealed specific patterns for each type of failure. For the HGB failure, the areas of highest activation were located in the sidebands around the fundamental frequency, which aligns with the theoretical basis describing this failure. These activations were consistently observed in all samples of this class. In the case of the BRB failure, activations were primarily concentrated at the fundamental frequency peak of 50 Hz, with additional minor activations in the sidebands. For the RBF failure, activations were distributed at the crest of the fundamental peak and dispersed around it. Finally, in the case of the BBF failure, two active regions were identified: one in the background band and another in the right sideband around 60 Hz, suggesting that the model focuses both on the position of the fundamental frequency and its corresponding sideband. These patterns were consistently repeated in all images of the test set for each respective class, indicating that the model identifies relevant features coherently. However, this analysis is limited to the direct relationship between the input image and the final prediction, without exploring the model’s internal activations in the intermediate layers.

To delve deeper into the analysis of the internal functioning of the model, Gradient-weighted Class Activation Mapping (GradCAM) was used to examine the activations in each convolutional layer of the VGG19 architecture. This architecture consists of 16 convolutional layers distributed across five feature extraction blocks, in addition to a sixth block dedicated to classification. GradCAM enables visualization of the regions in the input image that are most relevant to the model’s decision, focusing specifically on the convolutional layers. Unlike Saliency Maps, GradCAM provides more interpretable activation maps, which are based on the class gradients with respect to the feature maps of the selected layer. These gradients are used as weights to combine the feature maps, generating a heatmap that highlights the regions important for prediction.

The heatmaps generated for each convolution and class in the test set were observed. Figure 6 presents a schematic of the VGG19 architecture, including its 16 layers and six blocks. It also shows examples of the heatmaps, where the activation colors range from blue, indicating lower activation, to red, and up to yellow, representing maximum activation. The rows correspond to (A) HGB, (B) BRB, (C) RBF, and (D) BBF.

In the output of the first convolutional block (layer 2), the activations were directly concentrated on the line corresponding to the FFT for all classes, indicating that this block captures the features related to the signal. In block 2 (layer 4), the activations highlight the enhanced background, where red zones appeared in a grid pattern, while the signal remained blue, suggesting that this block primarily focuses on the background. As the convolutions progressed, the activations became more specific to each class.

For the HGB fault, the activations were localized in a lateral zone adjacent to the fundamental frequency in layer 9, indicating that this layer determines the class membership. In the case of the BRB fault, the activations concentrated on the two sidebands near the fundamental frequency, which was also observed in layer 13. For the RBF fault, the activations occurred in the final layer, surrounding the fundamental frequency with a pattern similar to that identified with previous explainability technique. Finally, in the case of the BBF fault, the activations were found in layer 14, standing out in the area under the curve of the fundamental frequency, with a slight shift to the right. In other cases, most layers did not show significant activation, except for some minor activations in specific samples that did not affect the classification probabilities.

In the case of the RBF fault, the activation occurred in the last layer of the network. As can be seen in the activation, the model did not focus on a specific point, but rather covered the entire area around the fundamental frequency. This behavior contrasts with the literature, where the detection of this fault has mainly been associated with specific harmonics. However, it is important to note that the model does not interpret the images in the same way as a human would, but rather according to the patterns it was able to adjust during training. Therefore, the activation in the last layer is considered to reflect a process of discarding the other three classes, where, by discarding these, the model finally concludes that the signal belongs to this class. This external and internal analysis of the model indicates that the model focuses on the behavior of the signal, the enhanced background, and also activates specific layers depending on the type of fault, highlighting patterns around the fundamental frequency.

### 4.4. Computational Resources

The model training was conducted using Google Colab, a cloud platform providing free computational resources, including GPU-compatible environments. Although this solution enabled efficient model training, the usage time limitations associated with free accounts required careful planning to complete the training tasks. PyTorch and fastai were used for model development, leveraging fastai’s simplicity for initial training and PyTorch’s flexibility for feature map analysis and explainability techniques.

The VGG19 model, selected for its performance, has a relatively low computational cost during the operational stage, consuming only 2 GB of GPU memory during inference. This makes it suitable for deployment in industrial environments where real-time operation is required. The training phase of the model was performed using an NVIDIA T4 GPU with 12 GB of memory on Google Colab, ensuring efficient learning despite the computational demands of deep learning. Once trained, the model can be deployed on devices with more modest hardware, such as modern laptops equipped with GPUs of at least 2 GB. Furthermore, the preprocessing step, which involves the signal-to-image transformation, can be performed on a CPU and requires no GPU resources. This combination of low operational cost and minimal preprocessing requirements allow the model to be a viable solution for both production systems and resource-constrained environments.

Regarding the data, the original dataset had a size of 3 GB due to the storage of raw signals. However, after preprocessing to convert them into spectral images, the dataset size was reduced to 320 MB, simplifying model handling and training. This aspect could be a topic of discussion in an industrial setting, as reducing data size lowers the costs associated with storage and processing. However, a production implementation would require an evaluation of the balance between computational resources and desired accuracy, especially when handling larger volumes of data in real time.

### 4.5. Limitations and Future Work

This study presents promising results in the automatic detection of faults in induction motors; however, certain limitations must be acknowledged. Firstly, the dataset used in this work was obtained from a controlled test bench under clean signal conditions. For real-world applications, it would be necessary to ensure the same level of signal clarity, as noise and interference commonly present in industrial environments could reduce the system’s performance. Future studies should address the impact of noise and explore preprocessing techniques to mitigate these effects.

Another limitation lies in the continuous evolution of deep learning architectures. While the VGG19 model demonstrated excellent performance in this study, future advances in neural network design may yield architectures capable of surpassing its results. Continuous evaluation and adaptation of the methodology to incorporate newer models will be essential.

From an implementation perspective, deploying the proposed system in industrial settings would require a dedicated computing infrastructure for each induction motor, including a GPU-enabled system. This approach entails a significant investment, making it less feasible for large-scale deployment. As a potential solution, future work will focus on designing a centralized model where data from multiple induction motors can be processed by a single system, reducing costs and simplifying integration.

In terms of fault coverage, this study focused on four specific types of faults. However, other types of faults exist that were not considered in this work. Expanding the dataset and training the model to identify a broader range of faults will be an important avenue for future research.

On the other hand, the FFT was chosen in this study due to its computational simplicity and its ability to capture spectral patterns under constant conditions. To work with variable conditions, future work will explore the use of the Short-Time Fourier Transform that is Gaussian Window Optimized [34] for the transient analysis or other transforms such as the Wavelet transform [67]. This approach would be useful in scenarios where the motor current signals present significant variability.

Finally, the study did not include healthy signal data, which are generally easier to classify. However, challenges arose when testing with a system exhibiting a single broken rotor bar, where the model struggled to differentiate the fault from a healthy state. This limitation will be addressed in a forthcoming manuscript, where we propose a solution to improve the model’s ability to handle these cases effectively.

## 5. Conclusions

The present work combines two contributions that enhance fault diagnosis in induction motors. First, the implementation of a differentiating background optimized the spectral representation, improving the extraction of relevant features and significantly increasing model accuracy. Second, the integration of local explainability techniques allowed for the interpretation of the model’s decisions, providing greater transparency and confidence in its use. These advancements were complemented by an exploration of the model’s internal behavior using techniques such as Saliency Maps and GradCAM, enabling the identification of how the overall image and each layer of the architecture contribute to fault classification.

Additionally, this work demonstrated the model’s ability to generalize across different operating conditions of load and speed, showcasing its consistency in fault detection. Although four specific types of faults were explored, the results obtained lay the groundwork for extending this approach to a broader range of anomalies, contributing to the development of more effective predictive maintenance systems. The combination of accuracy, interpretability, and low computational cost positions the proposed approach as a promising solution for industrial applications. However, pending challenges were identified, which will be addressed in future work.

## Figures and Tables

**Figure 1 sensors-25-00471-f001:**
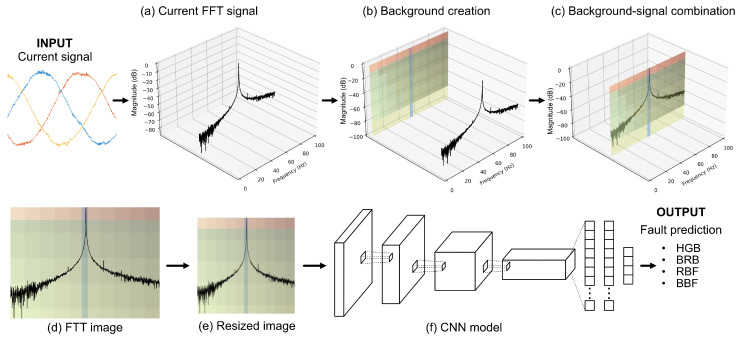
Overview of the background-enhanced FFT signal processing for fault detection. In (**a**), the original current FFT signal is displayed. In (**b**), the distinctive background is created. In (**c**), the background is combined with the FFT signal, forming a unified representation. (**d**) shows the final background-enhanced FFT image. In (**e**), the image is resized to a 224×224 pixels. Finally, (**f**) represents the input to a CNN model for automatic fault detection.

**Figure 2 sensors-25-00471-f002:**
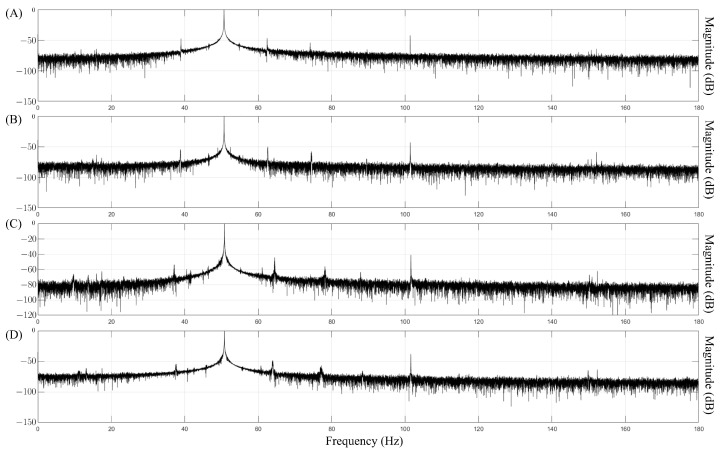
Fourier spectra of the current signals for each fault type at 1500 rpm and 100% load. (**A**) HGB, (**B**) BRB, (**C**) RBF, and (**D**) BBF. The x axis represents the frequency in Hz (up to 180 Hz for visualization), and the y axis represents the magnitude in decibels (dB). The model analyzes frequencies up to 100 Hz.

**Figure 3 sensors-25-00471-f003:**
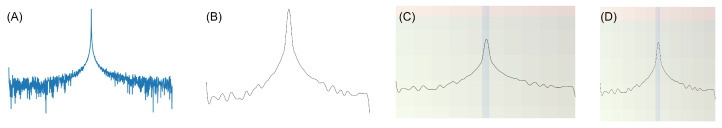
Transformation process of the FFT image. (**A**) Original FFT signal. (**B**) Smoothed signal after applying the Savitzky–Golay filter of degree 3. (**C**) Smoothed signal with a degraded background, including horizontal and vertical reference stripes. (**D**) Final resized image (224 × 224 pixels) prepared for input to the CNN model.

**Figure 4 sensors-25-00471-f004:**
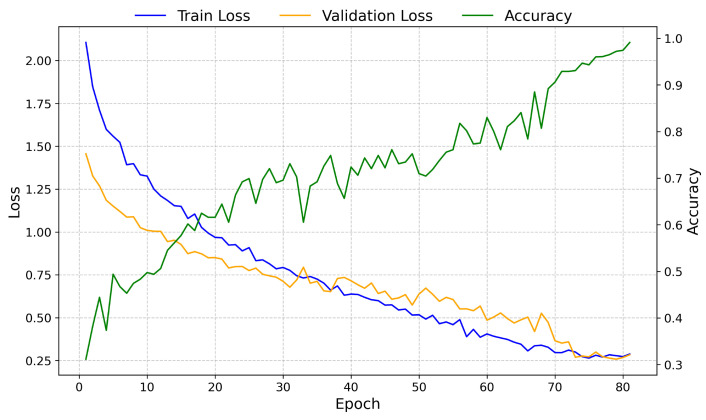
Training and validation loss, along with training accuracy, over 81 epochs. The loss curves indicate the convergence behavior of the model, while the accuracy curve indicates performance improvements, reaching a validation accuracy of 0.991 at the final epoch.

**Figure 5 sensors-25-00471-f005:**
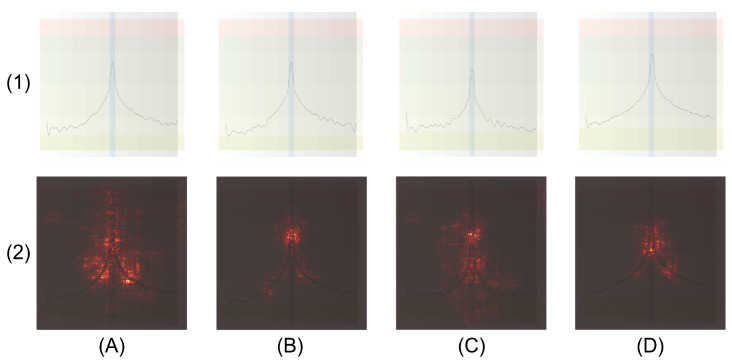
Images corresponding to the failure classes (**A**) HGB, (**B**) BRB, (**C**) RBF, and (**D**) BBF obtained under operating conditions of 1500 rpm and 100% load (row 1). Row 2 presents the saliency maps generated for each image, highlighting the relevant areas used by the model to perform the classification.

**Figure 6 sensors-25-00471-f006:**
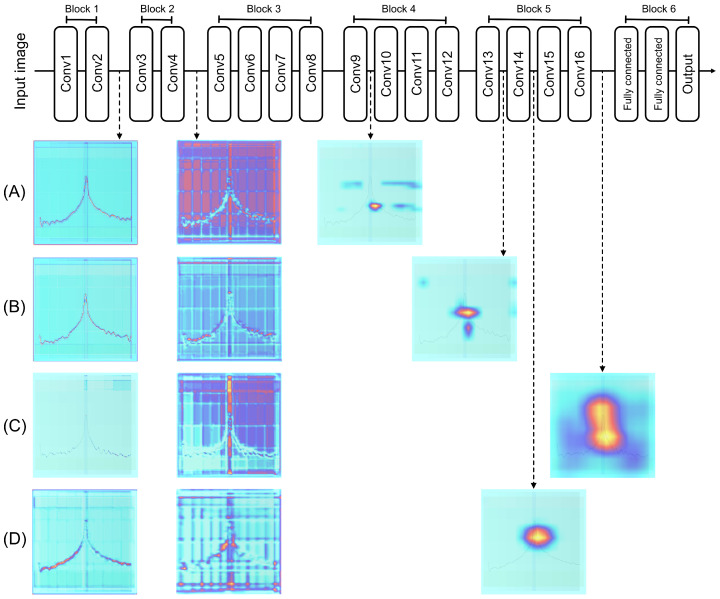
Visualization of the model’s internal interpretability using GradCAM in the VGG19 architecture, which consists of 16 convolutional layers distributed across 6 blocks (the sixth corresponds to classification). The activation maps generated for the classes (**A**) HGB, (**B**) BRB, (**C**) RBF, and (**D**) BBF highlight the regions relevant for prediction, where blue represents lower activation, red intermediate activation, and yellow maximum activation.

**Table 1 sensors-25-00471-t001:** Distribution of samples per class before and after balancing using Min-Max method. Image sets are divided into training, validation, and test sets.

Data Set	HGB	BRB	RBF	BBF	Total
Training	229	301	134	144	808
Validation	61	76	34	36	207
Testing	70	94	42	45	251
Balanced training	202	202	202	202	808
Balanced validation	51	51	51	51	204

**Table 2 sensors-25-00471-t002:** Comparative performance metrics for tested architectures. The table includes test set accuracy, training time, GPU resource usage in MiB, and the number of training epochs stopped by the early stopping technique.

Architecture	Accuracy (%)	Training Time (Min)	GPU Resource (MiB)	Training Epochs
VGG19	98.4	192	2760 MiB	81
VGG16	96.8	168	2443 MiB	74
ResNet154	97.1	270	3017 MiB	95
ResNet50	92.9	126	1722 MiB	65
InceptionV4	90.2	228	2329 MiB	89
SENet154	94.3	312	3671 MiB	101
EfficientNetV2	94.3	144	4858 MiB	56
ConvNext	93.9	216	3191 MiB	73
ViT-Large	96.8	384	7211 MiB	120
ViT-Base	93.9	282	5487 MiB	92
ViT-Small	92.9	252	3502 MiB	78

**Table 3 sensors-25-00471-t003:** Structure of the VGG19 architecture, including layers, type of operation performed, and output form.

Layer Type	Convolution	Output Shape
Input layer	-	224×224×3
Convolutional (3 × 3, ReLU)	Conv1, Conv2	224×224×64
Max pooling (2 × 2)	-	112×112×64
Convolutional (3 × 3, ReLU)	Conv3, Conv4	112×112×128
Max pooling (2 × 2)	-	56×56×128
Convolutional (3 × 3, ReLU)	Conv5, Conv6, Conv7, Conv8	56×56×256
Max pooling (2 × 2)	-	28×28×256
Convolutional (3 × 3, ReLU)	Conv9, Conv10, Conv11, Conv12	28×28×512
Max pooling (2 × 2)	-	14×14×512
Convolutional (3 × 3, ReLU)	Conv13, Conv14, Conv15, Conv16	14×14×512
Max pooling (2 × 2)	-	7×7×512
Flatten layer	-	25,088
Fully connected (ReLU)	-	4096, 2048
Output layer (Softmax)	-	4

**Table 4 sensors-25-00471-t004:** Confusion matrix for the classification of the four faults in the test set. Rows represent the true classes and columns the predictions. Values are presented normalized and in number of images in parentheses.

		Predicted Class			
		**HGB**	**BRB**	**RBF**	**BBF**
True Class	Healthy motor coupled to a generator with a broken bar (HGB)	0.99 (69)	0.01 (1)	0	0
	Broken rotor bar (BRB)	0	1.00 (94)	0	0
	Race bearing fault (RBF)	0	0	1.00 (42)	0
	Bearing ball fault (BBF)	0	0.02 (1)	0.03 (1)	0.95 (43)

**Table 5 sensors-25-00471-t005:** Summary of the model’s classification performance metrics for each fault class, including precision, recall, F1-score, and Jaccard Index, as well as global averages.

Faults	Precision	Recall	F1-Score	Jaccard Index
HGB	1.000	0.990	0.995	0.990
BRB	0.971	1.000	0.985	0.971
RBF	0.971	1.000	0.985	0.971
BBF	1.000	0.950	0.974	0.950
Average	0.986	0.985	0.985	0.971

**Table 6 sensors-25-00471-t006:** Classification accuracy of the model across various load conditions. Each value represents the overall accuracy for a specific fault type at the indicated load percentage.

Load Condition	HGB	BRB	RBF	BBF
100%	1.00	1.00	1.00	1.00
75%	1.00	1.00	1.00	1.00
50%	1.00	1.00	1.00	0.96
25%	0.94	1.00	1.00	1.00
0%	1.00	1.00	1.00	1.00

**Table 7 sensors-25-00471-t007:** Classification accuracy of the model across different motor speeds. Each value reflects the accuracy for detecting a specific fault type at the corresponding operating speed.

Speed (rpm)	HGB	BRB	RBF	BBF
1500	1.00	1.00	1.00	0.98
1200	1.00	1.00	1.00	1.00
900	0.96	1.00	1.00	1.00
20	1.00	1.00	1.00	1.00

## Data Availability

Data are contained within the article.

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
