# Peer review of "Fault Detection in Induction Machines Using Learning Models and Fourier Spectrum Image Analysis"

_sensors, 2025, doi:10.3390/s25020471_

Round 1

Reviewer 1 Report

Comments and Suggestions for Authors

I could not get *maybe it was presented somewhere the following issues, so maybe you can present it somewhat better:

a) what is the signal length used for the FFT

b) more important: How did you do the segementation, and perhaps, was there any influence on the length of the segmentation. Some windowing (not just zero-padding) would be my advice.

c) did the experimental setup allow for variable speed or only devices running at constant speed (as I understand it).

d) Have you tried to compare the resuls to the simple inspection of a spectrogram, i.e. the usual (low complexity) short-time Fourier transform with say a broad Gaussian window?

e) Can you say more about the "computational load" . Going to image processing tasks seems to be cumbersome. Can the algorithm be used on a laptop (after training maybe on a larger machine)?

f) How did you implement the  Savitzky-Golay filter? It is a real FIR filter involving entries from the inverse of a (perhaps partial) Vandermonde matrix, so it can be implemented using an FFT based algorithm instead of a coordinatewise polynomial fit! How is it done and can you explain it better?

Reviewer 2 Report

Comments and Suggestions for Authors

The authors of the paper propose a method for automatic fault detection in induction motors based on signal processing and deep learning techniques. In order to consider the article for publication, it is necessary to review and clarify certain aspects: 

1. It is necessary to clarify why the authors consider that the addition of background is a contribution in the category of imaging techniques

2. It is necessary to explain in Section 2.2 what procedures were used in the data collection to implement or simulate the failures

3. The Min-Max scaling technique transforms the features by scaling each of them to a given range. However, the authors claim to have used it for data balancing. This should be reviewed and argued or corrected

4. Since the images of the dataset are generated in a controlled way by an algorithm (including the addition of background), it is necessary to explain in more detail why the use of data augmentation is justified

5. In Table 2 it is advisable to add the number of training epochs, the size of the model in MB and whether the training time involved the use of GPU

6. Before section 4.2 it is not clear in the paper whether transfer learning was used or not. For the sake of clarity in the document, I recommend that when presenting the architecture used (VGG19), it should be stated that it was trained from scratch. Also, specify in Table 2 whether these results correspond to training from scratch or by transfer learning

7. In the first paragraph of Section 4.3, correct the phrase “interpretability techniques were proposed”, since the paper does not propose these techniques, but uses them

8. As discussed until the end of the paper, the method should be oriented towards a practical model of anomaly detection in routine operation (i.e. including healthy data) and not only of fault classification. Hence, it is necessary to include at the beginning of the paper the justification of the applicability of the proposed method.

Round 2

Reviewer 2 Report

Comments and Suggestions for Authors

The manuscript has been improved according to the observations of the first revision. 

I suggest an additional adjustment:

Replace “adjusted” with “fine-tuned” in the sentence “Transfer learning was used, where the pre-trained weights of the model are used and these are adjusted with the data set described ”.